# Crocodilepox Virus Protein 157 Is an Independently Evolved Inhibitor of Protein Kinase R

**DOI:** 10.3390/v14071564

**Published:** 2022-07-19

**Authors:** M. Julhasur Rahman, Loubna Tazi, Sherry L. Haller, Stefan Rothenburg

**Affiliations:** 1Department of Medial Microbiology and Immunology, School of Medicine, University of California Davis, Davis, CA 95616, USA; rahman.mohammad-julhasur@gene.com (M.J.R.); ltazi@ucdavis.edu (L.T.); 2Purification Development Department, Genentech, Inc., One DNA Way, South San Francisco, CA 94080, USA; 3Center for Biodefense and Emerging Infectious Diseases, University of Texas Medical Branch at Galveston, Galveston, TX 77555, USA; shhaller@utmb.edu

**Keywords:** poxviruses, protein kinase R, evolution, translational regulation, eIF2

## Abstract

Crocodilepox virus (CRV) belongs to the *Poxviridae* family and mainly infects hatchling and juvenile Nile crocodiles. Most poxviruses encode inhibitors of the host antiviral protein kinase R (PKR), which is activated by viral double-stranded (ds) RNA formed during virus replication, resulting in the phosphorylation of eIF2α and the subsequent shutdown of general mRNA translation. Because CRV lacks orthologs of known poxviral PKR inhibitors, we experimentally characterized one candidate (CRV157), which contains a predicted dsRNA-binding domain. Bioinformatic analyses indicated that CRV157 evolved independently from other poxvirus PKR inhibitors. CRV157 bound to dsRNA, co-localized with PKR in the cytosol, and inhibited PKR from various species. To analyze whether CRV157 could inhibit PKR in the context of a poxvirus infection, we constructed recombinant vaccinia virus strains that contain either CRV157, or a mutant CRV157 deficient in dsRNA binding in a strain that lacks PKR inhibitors. The presence of wild-type CRV157 rescued vaccinia virus replication, while the CRV157 mutant did not. The ability of CRV157 to inhibit PKR correlated with virus replication and eIF2α phosphorylation. The independent evolution of CRV157 demonstrates that poxvirus PKR inhibitors evolved from a diverse set of ancestral genes in an example of convergent evolution.

## 1. Introduction

The members of the *Poxviridae* family are large double-stranded DNA viruses, which exclusively replicate in the host cytoplasm. The genomes of poxviruses range between 134 kb and 360 kb, and encode approximately 200 genes [1]. Poxvirus members can infect vertebrates and invertebrates, collectively exhibiting a broad host range. Whereas some members strictly infect one or a limited number of host species, others can infect a large number of host species. For example, variola virus (VARV), the causative agent of smallpox disease, only productively infects humans. In contrast, other poxviruses, including cowpox viruses (CPXV), monkeypox virus (MPXV), and vaccinia virus (VACV), can infect a wide range of host species [2,3,4].

Poxvirus infections have been reported worldwide in host species from the *Crocodylia* order, including caimans (Spectacled and Brazilian caimans) and crocodiles (Nile, saltwater, and freshwater crocodiles), especially in farmed caimans and crocodiles [5,6,7,8,9]. So far, two poxviruses infecting *Crocodylia* have been completely sequenced and assigned to the *Crocodilydpoxvirus* genus of the *Chordopoxvirinae* subfamily: crocodilepox virus (CRV), isolated from farmed Nile crocodiles [10], and saltwater crocodilepox virus (SwCRV), isolated from Australian saltwater crocodiles [11]. CRV mainly infects hatchling and juvenile Nile crocodiles and causes scattered, circular, grey-to-white skin lesions on the dorsal and ventral parts of the body, including the hind feet, mouth cavity, eyes, nostrils, head, and tail regions [8,12,13]. Lesions on the eyes and head can impair vision and lead to skin deformities. These skin lesions can lead to further opportunistic infections by *Dermatophilus*-like bacteria or fungi. Although the mortality rate is, in general, low in Nile crocodiles, the morbidity rate and economic impact of the virus in farmed crocodiles is high [12,14].

Morphologically, CRV and SwCRV virions are similar to virions of orthopoxviruses, the most widely studied genus of *Chordopoxvirinae*. The virions are brick-shaped and contain a dumbbell-shaped central core, which is flanked by lateral bodies [6,15,16]. The genome of CRV is 190,054 bp long and, unlike orthopoxviruses, has a high GC content (62%). A total of 173 protein-coding genes have been predicated in the CRV genome, which contains two short (1754 bp) inverted terminal repeats (ITRs). An interesting feature of CRV is that its genome lacks recognizable homologs of most host-immune evasion genes found in other chordopoxviruses, suggesting that it evolved unique genes to manipulate the antiviral host response. CRV encodes 62 predicted proteins that do not show apparent sequence homology with non-*Crocodylidpoxvirus* proteins. Forty-eight of these protein-encoding genes are located in the terminal regions [10].

The ability of poxviruses to enter host cells is independent of species-specific cellular receptors; poxviruses are therefore able to enter into many different cell types [2,17]. Because entry is not a barrier, the successful replication of poxviruses depends on their ability to subvert host immune responses [2,3]. During virus infections, host cells induce type I interferons (IFNs) and other antiviral host pathways, leading to the establishment of an antiviral state [18]. Poxviruses accommodate a variety of different immunomodulatory genes, which interfere with host antiviral proteins to allow virus replication [19]. One of these IFN-induced host antiviral proteins is protein kinase R (PKR), which is targeted by many viruses, including poxviruses [20,21,22,23,24]. PKR is present at intermediate levels in unstimulated cells as an inactive monomer. Upon the detection and binding of double-stranded (ds) RNA produced during virus replication and transcription, PKR undergoes conformational changes, which lead to PKR activation via homodimerization and autophosphorylation. Activated PKR phosphorylates the alpha subunit of eukaryotic translation initiation factor 2 (eIF2), which leads to the suppression of general mRNA translation and, in turn, restricts virus replication [25,26,27,28,29,30,31,32]. In addition, PKR can also inhibit virus replication by inducing proinflammatory responses and cell death in infected cells [33,34,35,36]. Most mammalian poxviruses inhibit PKR by encoding orthologs of two vaccinia virus PKR inhibitors: E3 and K3 [37,38]. E3 inhibits PKR activation by binding to dsRNA and inhibiting PKR homodimerization [22,39,40,41]. K3 is a pseudosubstrate inhibitor of PKR, which restricts eIF2α-PKR interactions [21,23,42].

CRV lacks homologs of known PKR inhibitors [10]. It is therefore not clear how it inhibits the PKR pathway. To investigate this, we performed BLAST database searches and identified a putative viral PKR inhibitor from open reading frame (ORF) 157 of the CRV genome, CRV157. CRV157 was previously characterized in silico as a putative dsRNA binding protein [43]. CRV157 contains a putative dsRNA binding domain (dsRBD) and an uncharacterized C-terminal domain that does not show homology to other known proteins. In this study, we investigated the hypothesis that CRV157, with its putative dsRNA binding domain, might act as a PKR antagonist. The results shown here indicate that CRV157 can bind dsRNA and inhibit PKRs from different species, and that its dsRNA binding ability is essential for PKR inhibition. Furthermore, CRV157 was able to rescue replication of a recombinant VACV that lacks both PKR inhibitors E3L and K3L. These results indicate that CRV is an independently evolved PKR inhibitor.

## 2. Materials and Methods

### 2.1. Cell Lines

European rabbit RK13 (kindly provided by Dr. Bernard Moss), HeLa (ATCC#CCL-22), HeLa-PKR^kd^ (knockdown) (kindly provided by Dr. Charles Samuel) [35], and RK13+E3+K3 (RK13 cells that stably express VACV E3L and K3L) [44] cell lines were maintained in Dulbecco’s Modified Essential Medium (DMEM, Life Technologies, Carlsbad, CA, USA) supplemented with 5% fetal bovine serum (FBS, Fischer Scientific, Waltham, MA, USA) and 25 µg/mL gentamycin (Quality Biologicals, Gaithersburg, MD). RK13+E3+K3 cells were additionally supplemented with 500 µg/mL geneticin (Life Technologies) and 300 µg/mL zeocin (Life Technologies). All cells were incubated at 37 °C, 5% CO_2_.

### 2.2. Plasmids

ORFs of all PKRs from the indicated species and viral genes were cloned into the pSG5 vector (Agilent, Santa Clara, CA, USA) under the control of a SV40 promoter for transient expression. The construction of knockdown-resistant human PKR, mouse PKR, European rabbit PKR, and Chinese hamster PKR was previously reported [45,46]. The chicken PKR (identical to XM_015283611.2) gene was cloned from a 14-day-old chicken embryo (*Gallus gallus*). Armenian hamster PKR and pig PKR were cloned from AHL-1 cells (ATCC-CCL-195) and swine PK15 cells (ATCC-CCL-33), respectively.

CRV157 (GeneID: 4363410) was obtained from gBlocks^®^Gene fragments (IDT) and cloned into the TOPO-TA vector (Invitrogen). The CRV157 ORF was then subcloned into the pSG5 vector with *Sac* I and *Xho* I restriction sites. CRV157-K48A/K49A, CRV157 delC, and CRV157 delN plasmids were generated by site-directed mutagenesis or deletion mutation PCR using pfu high-fidelity DNA polymerase (Life Technologies, Carlsbad, CA, USA). For site-directed mutagenesis, PCR was performed using pSG5-CRV157 as a template with the primers containing lysine to alanine mutations (AAA to GCA and AAG to GCG) from nucleotide position 142 to 146. For the C-terminal deletion, nucleotide positions 1 to 198 were PCR amplified and cloned into pSG5 vector; for the N-terminal deletion, nucleotide positions 199 to 384 were PCR amplified and cloned into pSG5 vector. To make FLAG epitope tags at the C-termini of VACV E3L, CRV157, CRV157-K48A/K49A, CRV157 del C, and CRV157 del N plasmids, these ORFs were cloned into a pSG5 vector containing two in-frame FLAG tag sequences. CRV157-mCherry and chicken PKR-EGFP fusion genes were constructed by fusion PCR, and the constructs were cloned into the pSG5 vector. All plasmid constructs were sequenced to confirm the correct sequences.

### 2.3. Transfection and Luciferase-Based Reporter (LBR) Assays

Luciferase assays were performed following the protocol described previously [45]. Briefly, HeLa-PKR^kd^ cells (5 × 10^4^ cells/well) were seeded in 24-well plates a day before the transfection. For each transfection reaction mix, 0.05 µg of firefly luciferase (Promega) and 0.2 µg of each of the indicated PKR plasmids were co-transfected with 0.8 µg of either E3L, CRV157, CRV157-K48A/K49A, CRV157 del C, or CRV157 del N plasmids using GenJet (SignaGen Laboratories, Frederick, MD), with transfection reagent to DNA ratios of 2:1. For the titration experiments, the indicated concentrations of CRV157 or CRV157 del C plasmids were co-transfected. Each transfection was performed in triplicate, and plasmid DNA concentrations for each transfection mix were kept constant by adding an appropriate amount of pSG5 vector plasmid where necessary. After 48 h, whole-cell lysates were extracted using mammalian lysis buffer (GoldBio, St Louis, MO), USA, followed by the addition of luciferin (Promega, Madison, WI, USA). Luciferase activity was determined in a Glowmax luminometer (Promega). Each experiment was conducted at least three times, and representative experiments are shown.

### 2.4. Viruses and Infection Assays

The Vaccinia Virus Copenhagen strain, VC-2, was used as a parental strain to produce all the recombinant viruses used in this study. VC-2 and its derivative vP872 (∆K3L) [21] were kindly provided by Dr. Bertram Jacobs. The construction of the vP872 derivative VC-R4 (∆E3L, ∆K3L, in which E3L was replaced by EGFP) was described previously [47]. Recombinant viruses including VC-R4-VACV E3L, VC-R4-CRV157, and VC-R4-CRV157-K48A/K49A were generated in the VC-R4 backbone by scarless integration of the indicated genes into the E3L locus of VC-R4, as described [47]. All the recombinant viruses were plaque purified three times. PCR amplification and the sequencing of recombinant virus integration sites were conducted to confirm the successful integration and correct sequencing of CRV157, VC-R4-CRV157-K48A/K49A, or E3L.

The replication kinetics of the VC-R4 recombinants were tested in the RK13 cell line. Briefly, RK13 cells (6 × 10^5^ cells) were plated in 6-well plates a day before the infection. RK13 cells were infected at a MOI of 0.01 with VC-R4, or VC-R4-derived recombinant viruses. Virus samples were sonicated before the infection (2 times for 15 s) and diluted in a 2.5% FBS-supplemented DMEM media. Each infection at each different time point was performed in duplicate. Growth medium from each well was aspirated to add virus inocula to the respective wells, and then incubated at 37 °C for 1 h. After a 1 h incubation period, virus inocula were removed. Each well was washed twice with phosphate-buffered saline (PBS), and fresh DMEM media, supplemented with 5% FBS, were added. Cells were then scraped and collected, along with the media, at 0, 12, 24, 48, 72, and 96 hpi. Collected lysates were then subjected to three rounds of freeze/thaw cycles followed by sonication (2 times for 30 s, 50% amplitude) in a cup sonicator (Qsonica Q500). The titration of the collected viruses was conducted in RK13+E3+K3 cells by measuring plaque-forming units per mL (pfu/mL).

### 2.5. Poly I:C Pull-Down Assays

Pull-down assays were performed following the protocol previously described [48]. Sepharose beads were prepared by dissolving one gram of lyophilized CNBr-activated sepharose powder beads (17-0430-01; GE Healthcare Life Sciences) in 40 mL of 1 mM HCl followed by mixing 10 times by end-to-end rotation. The beads were then pelleted by centrifuging at 1000× *g* for 5 min; the supernatant was then removed without disturbing the beads. The addition of 1 mM HCl and the removal of the supernatant after centrifugation was repeated 4 times. After the final wash, HCl was removed, and the beads were ready for Poly I:C conjugation. A stock solution of poly I:C was prepared in a coupling buffer (0.1 M NaHCO_3_, 0.5 M NaCl, pH 8.3) at a concentration of 10 mg/mL. One gram of prepared sepharose beads was mixed with 2 mL of poly I:C stock solution in a total volume of 4 mL in coupling buffer, and mixed by end-over-end rotation at RT for 2 h. Unconjugated poly I:C was removed by washing the beads 5 times with equal volumes of coupling buffer and centrifugation at 1000× *g* for 5 min between washes and removal of the supernatant. Beads were then incubated with 0.1 M Tris HCl, pH 8 for 2 h at RT to block the unconjugated active sites on the beads; this was followed by 3 cycles of washing of the conjugated beads with alternating low-pH (0.1 M sodium acetate, 0.5 M NaCl, pH 4.0) and high-pH (0.1 M Tris-HCl, 0.5 M NaCl, pH 8.0) buffer (5 volumes of bead volume of each pH buffer were used). After the final wash, the beads were stored in co-IP lysis buffer (20 mM Tris-HCl, pH 8, 137 mM NaCl, 10% glycerol, 1% Triton-X, 2 mM EDTA) before use as a 50% bead slurry (with an equal volume of buffer added to the beads).

HeLa-PKR^kd^ cells were transfected at 70–80% confluency in 6-well plates with 3 µg of the plasmids pSG5-FLAG, pSG5-E3L-FLAG, pSG5-CRV157-FLAG, or pSG5-CRV157-K48A/K49A-FLAG. After 48-h, the media were removed and washed with PBS buffer; this was followed by cell lysis with 300 µL of lysis buffer (20 mM Tris-HCl, pH 8, 137 mM NaCl, 10% glycerol, 1% Triton X-100, 2 mM EDTA, protease inhibitor) on ice. For the pre-pull-down assay, 30 µL of lysed samples were separated, and the rest of the samples were centrifuged at 5400× *g* for 10 min to remove cellular debris. Pull-down of poly I:C was conducted by adding 50 µL of poly I:C bead slurry to each cell lysate sample, which was followed by end-over-end rotation at RT for 2 h and subsequent centrifugation at 5400× *g* for 2 min and removal of the supernatant. The samples were then washed 3 times with 200 µL of lysis buffer; after the final wash, proteins were released from the poly I:C beads, and the samples were resuspended in 120 µL of Laemmli buffer (6× SDS, β-mercaptoethanol) and denatured at 95 °C for 10 min. The samples were then centrifuged at 5400× *g* for 2 min, and the supernatants were collected and subjected to Western blotting.

### 2.6. Western Blots

Protein lysates from HeLa-PKR^kd^ cells transfected with the indicated plasmid constructs (3 µg) were collected 48 h post-transfection in 1% sodium dodecyl sulfate (SDS) in PBS (VWR). Protein lysates from RK13 cells infected with either VC-R4 (∆E3L∆K3L), VC-2, VC-R4-VACV E3L, VC-R4-CRV157, or VCR2-CRV157-K48A/K49A at MOI = 3 were collected at 6 h post-infection (hpi) in 1% SDS. All protein lysates were sonicated (twice for 5 s at 50% amplitude) to shear genomic DNA, separated by 12% SDS-polyacrylamide gel electrophoresis, and blotted onto Hybond-PVDF (GE Healthcare, Chicago, IL, USA) membranes using a methanol-based wet transfer apparatus (BioRad, Hercules, CA, USA). Membranes were blocked for 1 h at room temperature with either SuperBlock blocking buffer (Fischer Scientific, Waltham, MA, USA) when detecting FLAG and ß-actin, 5% (wt/vol) BSA when detecting phosphorylated eIF2α, or 5% (wt/vol) nonfat milk when detecting total eIF2α in TBST buffer (20 mM Tris, 150 mM NaCl, and 0.1% Tween-20 pH 7.4). The blots were then incubated overnight at 4 °C with the appropriate primary antibody: either mouse anti-D (1:1000, FLAG, abm), rabbit eIF2α-P (1:1000, Santa Cruz Biotechnology, sc-101670), rabbit total eIF2α (1:1000, Santa Cruz Biotechnology, sc-11386), or mouse anti-β-actin (1:1000, Sigma-Aldrich, Burlington, MA) diluted in TBST buffer. The membranes were then washed three times for 5 min each in 1× TBST buffer; this was followed by incubation with horseradish peroxidase-conjugated secondary antibodies (donkey anti-rabbit-HRP or donkey anti-mouse-HRP, Life Technologies) in blocking buffer for 1 h at room temperature. The membranes were then washed 3 times 10 min each in TBST buffer. After washing the membranes, proteins bound by the secondary antibody were detected with Proto-Glo ECL (National Diagnostics, Atlanta, GA, USA), and images were taken with either a Kodak-4000 MM Image Station or iBright (Invitrogen).

### 2.7. Phylogenetic Analysis

The dsRBD protein sequences from 62 viral proteins were aligned using MUSCLE [49]. A maximum likelihood phylogenetic tree was then generated from the multiple sequence alignment using PhyML 3.0 with 100 bootstrap replicates [50]. The resulting tree was visualized using the program FigTree v1.4.3 (http://tree.bio.ed.ac.uk/software/figtree/ (accessed on 1 June 2018)).

### 2.8. Confocal Microscopy

To determine the localization of PKR and CRV-157, HeLa-PKR^kd^ cells were plated either on glass coverslips in 12-well plates or on 35 mm coverglass bottom dishes (MatTek, Ashland, MA). At 40–50% confluency, the cells were co-transfected with EGFP-tagged chicken PKR (0.5 µg), and mCherry tagged CRV157 (0.5 µg) and incubated at 37 °C for 48 h. After the incubation, the media were removed, and cells were washed with PBS. The cells were then fixed with 2% formaldehyde and incubated for 20–30 min, followed by PBS washing to remove the formaldehyde. Following formaldehyde removal, the cells were incubated with 50 mM glycine for 5–10 min at RT and then washed with PBS. The cells were then incubated with nuclear stain, DAPI (300 nM), for 15–20 min and washed (×3) with PBS. Glass coverslips were removed and placed upside down on glass slides with mounting media. Excess mounting media were absorbed from the edges using filter paper and the edges were sealed by applying nail polish. After nuclear staining, the cells on the glass bottom dish were kept in PBS. 

Samples were then observed with an inverted Carl Zeiss LSM 880 confocal microscope using excitation wavelengths of 405 nm (DAPI), 514 nm (EGFP), and 633 nm (mCherry) using a 40× EC Plan-Neofluar oil objective with a 1.3 numerical aperture (NA). Fluorescence was detected using a gallium arsenide phosphide (GaAsP) detector to ensure an improved signal to noise ratio. Fluorescent images were obtained by sequential scanning of each channel to ensure reliable quantification of colocalization. Images of random fields were acquired, and channels were merged in Zeiss Zen microscope software during acquisition, which allowed us to make qualitative judgments about the colocalization of CRV157-mCherry and chicken PKR-EGFP signals. Confocal images were then imported to Fiji (v2.00-rc-68/1.52f) [51] software for colocalization analysis. 

### 2.9. Quantitative Colocalization Analysis (QCA)

Images acquired from the confocal microscope were processed using Fiji and the JACoP plugin [51,52]. Briefly, the backgrounds of channels with mCherry and EGFP were subtracted from the control confocal images (no fluorescent/fluorescence) by using the image calculator tool. Furthermore, the background was corrected using Coste’s automatic threshold for all channels. Pearson’s correlation coefficients (PCC) were calculated using the Pearson’s coefficient tab of the JACoP plugin. PCC values range from −1.0 to 1.0, where −1.0 implies no overlap and 1.0 indicates complete colocalization [53]. The data obtained were analyzed in Excel.

## 3. Results

### 3.1. CRV157 Contains a dsRNA-Binding Domain

In order to identify putative PKR inhibitors in other poxviruses, we performed PSI-BLAST searches of poxvirus sequences using default parameters with homologs of E3 and K3 from various poxviruses. For example, when we used the dsRNA binding domain of VACV E3 or the racoonpox virus E3 homolog as bait, CRV157, a putative protein encoded by CRV, was identified in iteration 2 but stayed below the PSI-BLAST threshold of 0.005 through iteration 4, when no new sequences were identified above the threshold. This indicates that CRV157 might be a dsRNA binding protein, distantly related to E3 orthologs. CRV157 was previously identified as a putative dsRNA binding protein when CRV proteins of unknown functions were searched against the NCBI database using PSI-BLAST [43]. The authors concluded that CRV157 was not an ortholog of the E3 family. To follow up, we extended our search for E3 homologs to include all viral proteins in the database. While dsRBDs from dozens of non-poxviruses were detected after iterations 2 and 3, CRV157 was only detected after iteration 4, and its ortholog (ORF198) from saltwater crocodilepox virus (SwCRV) only after iteration 5. To study the relationships among viral dsRBDs, we performed a phylogenetic analysis from a multiple sequence alignment containing dsRBDs from various viruses. In this analysis, CRV157 and SwCRV198 were found in a different clade than E3 sequences from other poxviruses (Figure 1). The basal branches were not supported by high bootstrap values, a known problem for phylogenetic analyses of dsRNA-binding proteins, because only a few amino acids are conserved [54]. However, together with the PSI-BLAST searches, these analyses further support the notion that CRV157 does not share an immediate ancestor with the poxvirus E3 family.

When PSI-BLAST searches with the full CRV157 sequences of the whole NR database were performed with initial BLAST thresholds set to 1, various dsRBDs of ribonuclease III proteins from *Streptomyces* species were discovered during the second iteration. The detected homology was not restricted to the dsRBD but extended a further 45 amino acids into the C-terminus (Figure 2A). A multiple sequence alignment with other dsRBDs, including VACV E3 homologs myxoma virus 029 and sheeppox virus 34, showed a deletion between beta sheets 1 and 2 in CRV157, as previously reported [43]. Sequence identities calculated from the alignment including only the dsRBDs showed that CRV157 and SwCRV198 had highest sequence identity with S. *gossypiisoli* ribonuclease III (Figure 2B). Combined, these data indicate that CRV157 is not an ortholog of the poxvirus E3 family, but originated independently. 

### 3.2. CRV157 Is a dsRNA Binding Protein

CRV157 contains a deletion between beta sheets 1 and 2, also called region 2, which has been shown to be important for dsRNA binding in other dsRBDs [55]. Therefore, we experimentally tested its dsRNA-binding capability in a pull-down assay using the synthetic dsRNA analog polyinosinic:polycytidylic acid (poly I:C). We assessed the poly I:C pull-down of CRV157 and a mutant of CRV157 (CRV157-K48A/K49A) in which key lysine residues, which were shown to be essential for dsRNA binding in other dsRBDs, have been replaced by alanine residues [40,48,56,57,58]. In order to detect and pull down the proteins, we added C-terminal FLAG epitope tags to CRV157, CRV157-K48A/K49A, and E3L in the mammalian expression vector pSG5. For the pull-down assay, HeLa-PKR^kd^ cells were transiently transfected with either the pSG5 vector, VACV E3L, CRV157, or CRV157-K48A/K49A. Cell lysates were incubated with poly I:C-conjugated sepharose beads, and the bound proteins were subjected to Western blot analysis. While all three proteins were detected in the initial lysates, only E3 and CRV157 were detected in the pull-downs, but not CRV157-K48A/K49A (Figure 3). These results demonstrate that CRV157 is a dsRNA binding protein.

### 3.3. CRV157 Is a PKR Inhibitor

To test whether CRV157 can inhibit PKR, we used an established luciferase-based reporter (LBR) assay [45]. In this assay, plasmids expressing PKR, potential PKR inhibitors and a luciferase reporter are co-transfected into PKR-depleted or PKR-deficient cells. Luciferase activity is used as a proxy for translation, where low luciferase activities indicate high PKR activity, and high luciferase activities indicate low PKR activity. In this assay, PKR is activated during transient transfection, likely by dsRNA formed from overlapping RNA transcripts of the transfected plasmids [59]. We co-transfected either empty vector, mouse or chicken PKR with empty vector, E3L, CRV157, or CRV157-K48A/K49A. Co-transfection of both E3L and CRV157 resulted in increased luciferase activities, which is indicative of PKR inhibition, whereas CRV157-K48A/K49A had no effect (Figure 4A). To analyze the contribution of the two CRV157 domains to PKR inhibition, we generated FLAG-tagged constructs encoding either the N-terminal (dsRBD) or C-terminal domains (Figure 4B), and performed LBR assays with these and full-length constructs. The construct containing the dsRBD resulted in luciferase activity comparable to that of full-length CRV157, whereas the C-terminal domain did not affect PKR activity (Figure 4C). Western blots of transfected cell lysates showed comparable expression of CRV157 and the N-terminal domain, whereas no expression was observed for the C-terminal domain (Figure 4D). Co-transfection of increasing amounts of CRV157 and the CRV157 N-terminus with chicken PKR revealed that full-length CRV157 inhibited PKR more efficiently than the dsRBD alone (Figure 4E).

### 3.4. CRV157 Can Inhibit PKR from Multiple Species

Multiple studies have previously shown that PKR inhibitors from poxviruses can show species specific PKR inhibition [45,46,60,61,62]. To investigate whether CRV157 exhibits species-specific PKR inhibition, we performed the LBR assay with CRV157 or VACV E3L as control, and PKR from the following seven vertebrate species: human, mouse, European rabbit, pig, Armenian hamster, Chinese hamster or chicken (Figure 4 and Figure 5). The PKR gene from the Nile crocodile, the natural CRV host, has not yet been sequenced and was therefore unavailable for these experiments. All tested PKRs were inhibited by E3 and CRV157, but to different extents. Human and mouse PKR were only weakly to moderately inhibited, whereas chicken PKR was the most strongly inhibited. In all cases, E3 showed stronger inhibition than CRV157.

### 3.5. Colocalization of CRV157 with PKR

To investigate whether CRV157 colocalizes with PKR, we fluorescently labelled CRV157 and chicken PKR at the C-termini, with mCherry (CRV157-mCherry) or EGFP (chicken PKR-EGFP), respectively. We co-transfected HeLa-PKR^kd^ cells with CRV157-mCherry and chicken PKR-EGFP to analyze their localization in cells using confocal microscopy. Confocal images were taken for quantitative colocalization analysis (QCA), which measures the degree of spatial coincidence between CRV157-mCherry and chicken PKR-EGFP using Fiji ImageJ analysis software. The degree of co-localization of CRV157 and chicken PKR was calculated from at least 15 representative confocal images from three independent experiments and is shown as the average of the Pearson’s correlation coefficients (PCC). The PCC value (0.77) from QCA indicates a strong correlation for the colocalization of CRV157 and chicken PKR in the cell cytoplasm (Figure 6A) [63]. Representative images of the localizations are shown (Figure 6B).

### 3.6. CRV157 Can Rescue the Replication of an Attenuated VACV Strain That Lacks Its PKR Inhibitors

The best test to assess whether CRV157 is a biologically meaningful PKR inhibitor is to insert it into a VACV strain that lacks its PKR inhibitors and determine whether it can rescue virus replication. We therefore inserted either CRV157, CRV157-K48A/K49A, or VACV E3L (as a positive control) into the replication-incompetent VACV strain VC-R4, which lacks both E3L and K3L. All of the transgenes were integrated into VC-R4 by a method that allows the seamless integration of transgenes into the E3L locus, and is driven by the same endogenous E3L promoter [47]. To compare the replication efficiency of these viruses relative to VC-2 (all viruses tested here were derived from this strain), vP872 (∆K3L), and the parental VC-R4 (∆E3L∆K3L), we infected rabbit kidney RK13 cells with these viruses at MOI = 0.01 over the course of 96 h (Figure 7A). Cell lysates were collected at different time points and viruses were titered on the permissive RK13+E3+K3 cell line. The replication kinetics indicate that VC-R4 and VC-R4+CRV157-KK-AA were replication defective. VC-R4 containing E3L (VC-R4+E3L) replicated as well as VC-2 and vP872 at all time points, reaching a plateau after 48 h. VC-R4 containing CRV157 (VC-R4+CRV157) showed replication, but it lagged behind the replication of the E3L-containing viruses at 12, 24, and 48 h by about 10-fold. However, replication to the E3L-containing viruses was comparable, at 72 and 96 h. VC-R4+CRV157 replicated to titers more than four orders of magnitude higher than the VC-R4 and VC-R4:CRV157-KK-AA viruses after 72 and 96 h.

Since eIF2α phosphorylation more directly demonstrates the effects of PKR inhibitors, we also monitored eIF2α phosphorylation levels in virus-infected RK13 cells by Western blot analysis (Figure 7B). To analyze eIF2α phosphorylation, we infected RK13 cells with the above mentioned VACV strains at MOI = 3 for 6 h. In RK13 cells, VC-R4+CRV157-K48A/K49A infection induced high levels of eIF2α phosphorylation, comparable to that seen in VC-R4 infected cells; whereas, VC-2, vP872 and VC-R4+E3L infections led to low eIF2α phosphorylation levels, and intermediate eIF2α phosphorylation levels were observed in VC-R4+CRV157 infected cells. Thus, PKR sensitivity and virus replication correlated well with eIF2α phosphorylation levels in infected RK13 cells. 

## 4. Discussion

Poxviruses infecting crocodilians pose threats for both farm-raised and wild populations [12,14,16]. To date, the full genome sequences from two such poxviruses (CRV and SwCRV) have been described; these form a monophyletic clade and have been assigned to the *Crocodylidpoxvirus* genus [10,11]. In-depth genomic characterization of multiple SwCRV isolates shows strong evidence of recombination within the SwCRV clade [64]. To the best of our knowledge, however, no functional characterizations of any crocodylidpoxvirus proteins have been reported. Here, we show that CRV157 is a PKR inhibitor and highlight the importance of the role of the PKR pathway in crocodylidpoxvirus infections.

Together with an earlier study [43], all the evidence presented here indicates that CRV157 evolved independently from the other poxvirus PKR inhibitors. While it is hard to trace the evolutionary origins of individual dsRBDs because there are relatively few conserved amino acid residues across different dsRBDs [54], the extended homology in the C-terminus of CRV157 with *Streptomyces* sp. RNase III indicates that it might have been horizontally transferred from an unsampled bacterium. However, the currently available sequences do not allow us to infer the origin of CRV157 with high confidence. While most poxvirus proteins show higher sequence identities with eukaryotic proteins in bioinformatic analyses, some poxvirus proteins appear to be more similar to bacterial proteins, indicating possible transfer events between bacteria and viruses [65]. Indeed, several CRV proteins show higher sequence identities to bacterial proteins than to eukaryotic proteins [10]. One of these is CRV051, which shows the highest sequence identities (about 32%) with prokaryotic serine recombinases [66].

Although CRV157 evidently contains a dsRBD, as previously reported [43], it was important to determine whether it was actually functional, especially in light of a deletion between beta sheets 1 and 2 (region 2), which include amino acid residues that have been shown to be involved in dsRNA interactions in other dsRBDs [55]. Poly (I:C) pull-down of wild-type CRV157, but not of CRV157-K48A/K49A, showed that CRV157 is a dsRNA binding protein. This binding activity correlated with the inhibition of PKR, which was only inhibited by wild-type CRV157, but not by CRV157-K48A/K49A. Importantly, CRV157, but not CRV157-K48A/K49A, was able to rescue replication of VACV that was missing K3L and E3L genes in RK13 cells. CRV157-containing virus titers were about 10-fold lower than those of the E3L-containing viruses up to three days post-infection. This finding indicates that CRV157 inhibited rabbit PKR sub-optimally, at least in the context of VACV infection. This correlated with the intermediate eIF2α phosphorylation levels induced by VC-R4+CRV157, and the finding that CRV157 was not as potent in inhibiting PKR from all tested vertebrates as VACV E3 in the LBR assay. It is possible that PKR from the natural host, the Nile crocodile, might be more sensitive to CRV157 inhibition. However, we could not test this because the PKR gene from this species is not known. Alternatively, CRV157 might work sub-optimally in the context of VACV infection, or in the tested mammalian cells. It is also possible that the intermediate PKR inhibition might be associated with the longevity of the CRV infection in Nile crocodiles, which can often persist for many months [6,12]. Another possibility is that intermediate PKR inhibition may have unexpected consequences, such as the functional activation of the pro-inflammatory NF-kB pathway, as recently shown [36]. In this case, the limited inhibition of translation resulted in NF-kB activation and the translation of induced mRNA, whereas no PKR inhibition abolished translation of the induced mRNAs. Nevertheless, while it is unclear what impact CRV157 would have against crocodile PKR, this series of experiments demonstrates that CRV157 is capable of inhibiting PKR.

An interesting observation from the LBR assay was that the individual tested PKRs appeared to exhibit differential sensitivity to both CRV157 and E3, with human PKR being inhibited the least and chicken PKR being inhibited the most strongly, indicating that some PKRs might be more sensitive to viral dsRNA binding proteins than others. Notably, expression of the dsRBD of CRV157 alone (CRV157 del C) did not inhibit PKR as efficiently as full length CRV157, indicating that the uncharacterized C-terminal domain of CRV157 might be also involved in PKR inhibition. This is reminiscent of the situation found in VACV E3 and the variola virus E3 homolog, for which the N-terminal Zα domain was shown to be important for PKR inhibition in addition to the C-terminal dsRBD in yeast assays [41,67]. In the context of vaccinia virus infection, an N-terminally truncated E3L gene resulted in altered eIF2α phosphorylation levels at 6 h, but led to increased eIF2a phosphorylation at 9 and 12 h after infection [68]. These findings show that PKR inhibition can be influenced by domains adjacent to the dsRBD. While we demonstrated that CRV157 is an inhibitor of PKR, it might have additional functions, such as inhibiting other dsRNA-sensing host proteins, as has been demonstrated for vaccinia virus E3 [69]. 

## 5. Conclusions

Poxviruses have evolved multiple inhibitors of antiviral host proteins, often targeting various steps in antiviral pathways [19]. The critical role of PKR in inhibiting virus replication is highlighted by the fact that most mammalian poxviruses contain two inhibitors, named E3 and K3 in the vaccinia virus, which prevent either PKR activation or PKR interaction with its substrate IF2α. Notably, E3 and K3 orthologs are missing in avipoxviruses and crocodylidpoxviruses [37,38]. Avipoxviruses possess a protein with homology to the eIF2α-binding and protein phosphatase 1 (PP1)-binding region of the DNA damage-inducible protein 34 (GADD34). Similarly to GADD34, canarypox virus protein 231 was able to suppress PKR-toxicity in a yeast assay, in which it also promoted PKR-mediated eIF2α dephosphorylation [70]. The data presented here show that CRV157 is a fourth PKR inhibitor found in poxviruses, and thus provide a compelling example of convergent evolution.

Because of the negative impact of crocodilepox viruses on crocodile farming, an important goal would be to develop a vaccine for this industry. Since vaccinia virus with targeted deletion in the PKR inhibitor E3L has been shown to protect mice and rabbits from lethal challenges from ectromelia virus and rabbitpox virus, respectively [71,72], CRV or SwCRV strains with deletions in CRV157 or SwCRV198 might be promising candidates for such vaccines. 

## Figures and Tables

**Figure 1 viruses-14-01564-f001:**
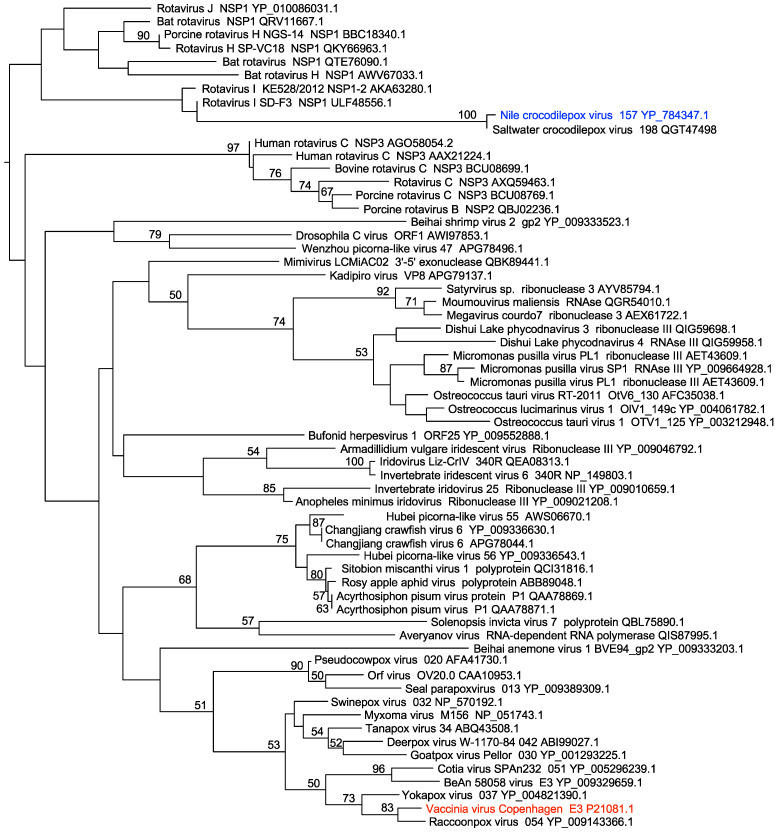
Phylogenetic relationship between viral dsRNA-binding domains. A maximum phylogenetic tree of dsRBDs from indicated viral proteins was constructed from a multiple sequence alignment generated with MUSCLE using PhyML. Numbers next to branches indicate bootstrap values ≥ 50. Accession numbers of proteins are indicated. The positions of CRV157 and VACV E3 are highlighted in blue and red, respectively.

**Figure 2 viruses-14-01564-f002:**
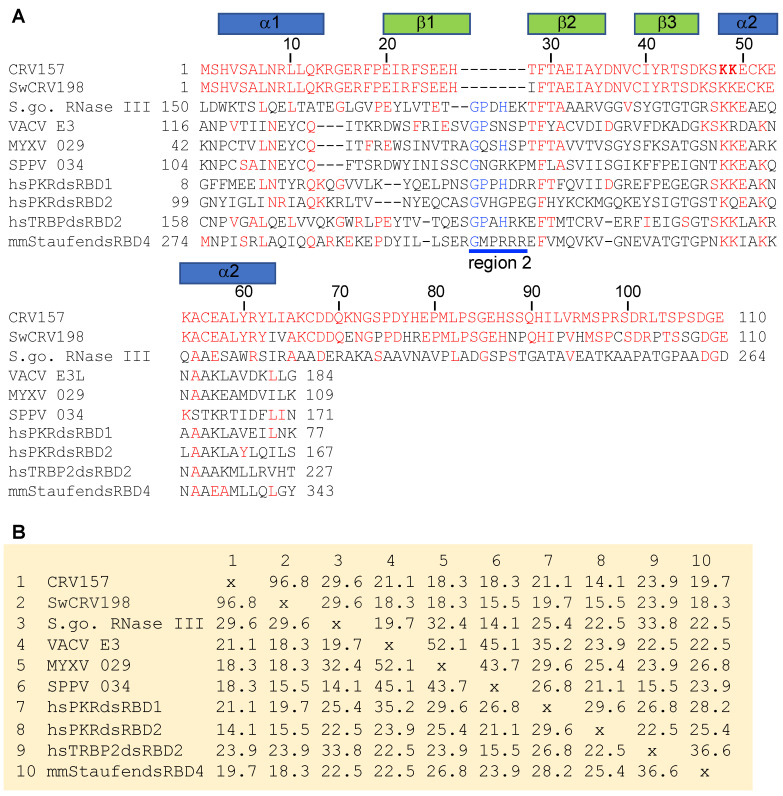
Multiple sequence alignment and sequence identities of CRV157, and selected double-stranded RNA-binding domains. (**A**) multiple sequence alignment of the amino acids from the following double-stranded RNA-binding domain containing proteins was created using MUSCLE: CRV157 (YP_784347.1); saltwater crocodilepox virus (SwCRV) ORF198 (QGT47498.1); *Streptomyces gossypiisoli* (S. go.) ribonuclease III (WP_232108110.1); vaccinia virus (VACV) E3 (AAA48040.1); myxoma virus (MYXV) 029 (NP_051743.1); sheeppox virus (SSPV) 034 (YP_001293225.1); homo sapiens (hs) PKR; hsTRBP (NP_599150.1; 5N8L_A (PMID: 29449323)); mus musculus (mm) Staufen homolog 2 (EDL14343.1). Secondary structures, as reported for mm Straufen (1UHZ_A), are indicated above the alignment. Sequences identical to CRV157 are shown in red. Sequences that were previously shown to be conserved in other dsRBDs in region 2 [55] are shown in blue. Two conserved lysins (K) in CRV157, which were replaced by alanine residues in this study, are shown in bold. (**B**) Percent identities between indicated dsRBDs from the above multiple sequence alignment were calculated in MegAlign.

**Figure 3 viruses-14-01564-f003:**
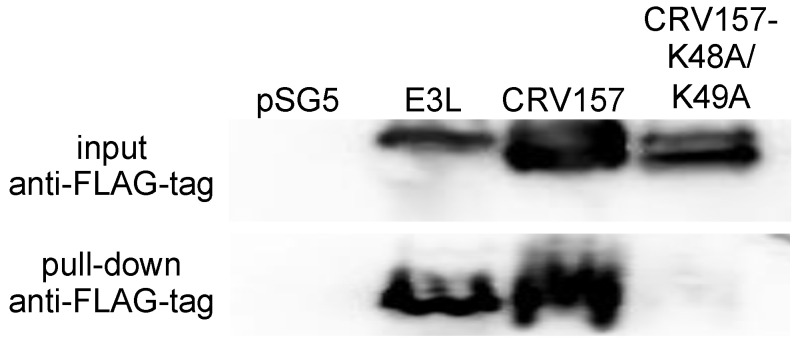
Pull-down of CRV157 and E3 with poly I:C. HeLa-PKR^kd^ cells were transfected with 3 μg of the indicated plasmids. Protein lysates were analyzed by Western blot before (input) and after pull-down with poly I:C sepharose with an anti-FLAG antibody.

**Figure 4 viruses-14-01564-f004:**
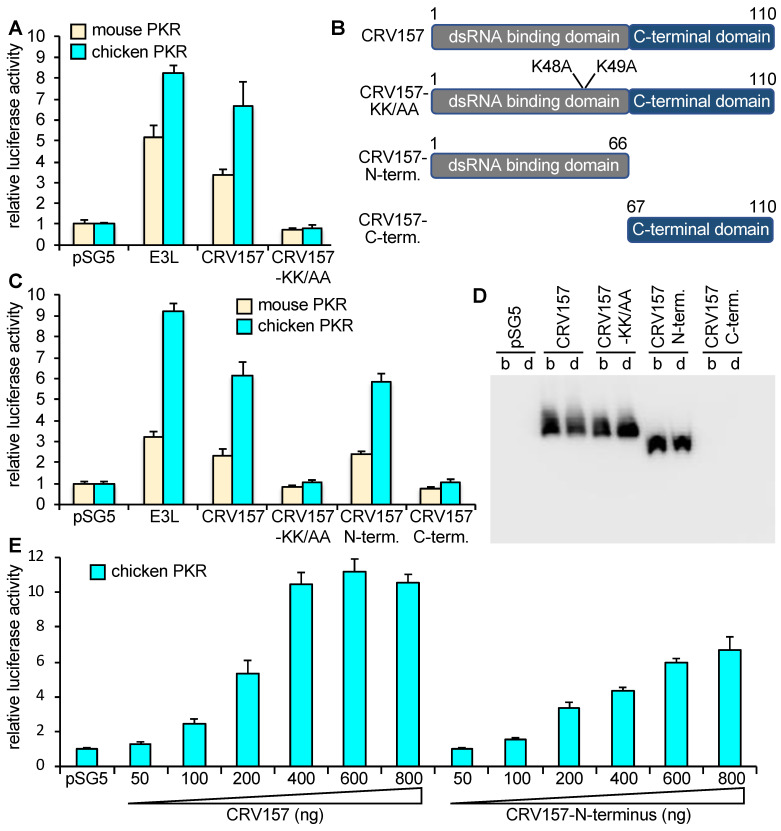
Inhibition of PKR by CRV157 and CRV157 mutants. (**A**) HeLa^PKRkd^ cells were transfected with the luciferase expression vector pGL3 (0.05 µg), PKRs (0.2 µg) from mouse or chicken, and 0.8 µg of either plasmid vector, VACV E3L, CRV157, or CRV157-KK-AA. Luciferase activities were measured 48 h after transfection and normalized to PKR-only transfected cells. The standard deviations shown were calculated from three independent transfections. All results shown are representative of at least three independent experiments. (**B**) Overview of CRV157 constructs used. (**C**) In addition to the plasmids mentioned under (A), HeLa^PKRkd^ cells were also transfected with CRV157-N-terminus (amino acids 1–66) or CRV157-C-terminus (amino acids 67–127) and assayed as described. (**D**) Western blot of total HeLa^PKRkd^ cell lysates transfected with indicated FLAG-tagged constructs. Lysates were either run with b-mercaptoethanol (b) or Dithiothreitol (DTT, d) as reducing agents. (**E**) HeLa^PKRkd^ cells were also co-transfected with increasing amounts of CRV157 or CRV156-N-terminus, along with chicken PKR 0.2 µg and pGL3 (0.05 µg). Luciferase activities were determined as described under A.

**Figure 5 viruses-14-01564-f005:**
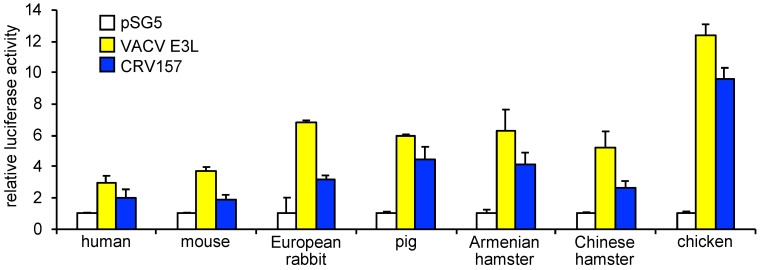
Species-independent PKR inhibition by CRV157. HeLa^PKRkd^ cells were co-transfected with firefly luciferase (0.05 µg) and indicated plasmids encoding mammalian PKRs (0.2 µg) with either CRV157 (0.8 µg) or VACV E3L (0.8 µg) plasmids. For both experiments, standard deviations from three independent transfections are shown. The results shown are representative of three independent experiments.

**Figure 6 viruses-14-01564-f006:**
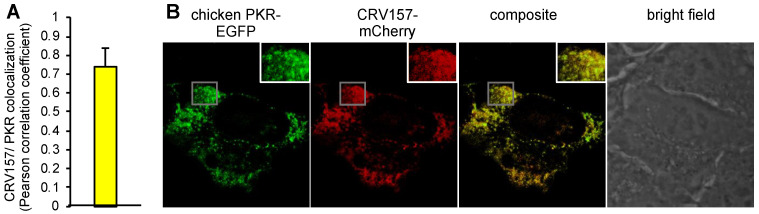
Co-localization of CRV157 with chicken PKR. (**A**) HeLa^PKRkd^ cells were co-transfected with plasmids expressing PKR-EGFP (0.5 µg) (green) and CRV157-mCherry (0.5 µg) (red). Fixed and permeabilized cells were subjected to confocal microscopy. (**A**) Results of quantitative colocalization analysis (QCA) indicating the degree of colocalization of CRV157 and chicken PKR. An average of the Pearson correlation coefficients (PCC) was calculated with 15 representative samples of three independent experiments. The error bar indicates the standard deviation. (**B**) Colocalization of CRV157 and chicken PKR is shown by the overlap of signals in the composite, which results in a yellow signal. Data shown are representative images from three independent experiments. The insets represent the boxed areas.

**Figure 7 viruses-14-01564-f007:**
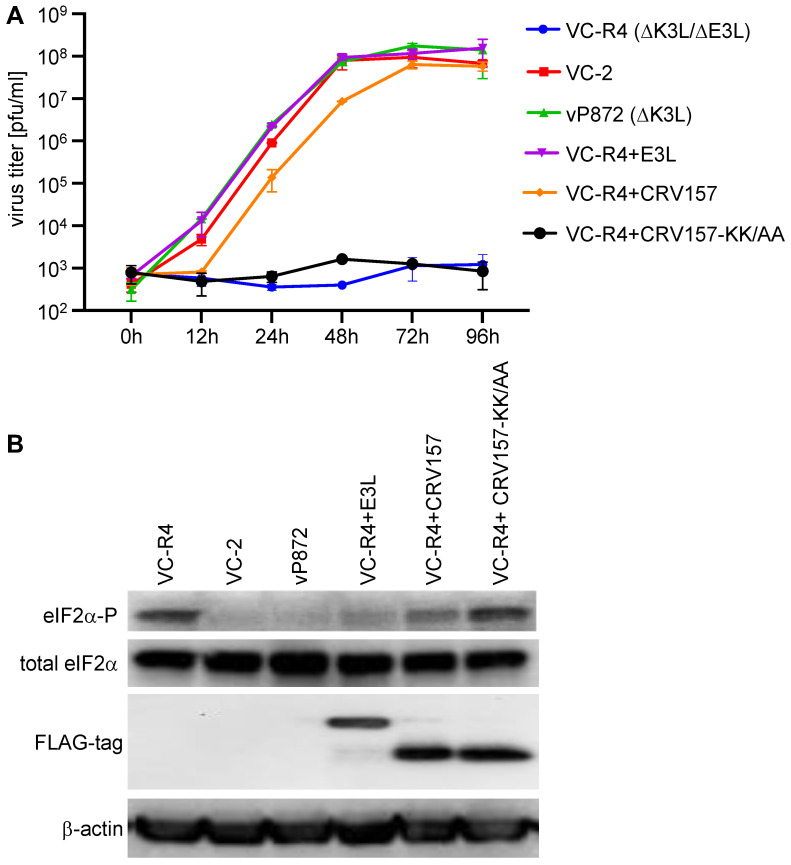
Rescue of VACV replication by CRV157 in RK13 cells. (**A**) RK13 cells were infected at MOI = 0.01 with wild-type VACV-Copenhagen (VC-2), vP872 (VACV ∆K3L), VC-R4 (VACV ∆E3L ∆K3L), or VC-R4 recombinants containing either E3L, CRV157, or CRV157-KK-AA. After the indicated time points, cell lysates were collected, and virus production was determined by titering on RK13+E3L+K3L cells. Standard deviations of two independent infections are shown. (**B**) Total protein lysates were collected from RK13 cells infected (MOI = 3) with the indicated viruses 6 h post-infection, and analyzed by Western blot analysis with antibodies specific for phosphorylated (P) or total eIF2α, FLAG tag, or β-actin.

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
