# Peer review of "Crocodilepox Virus Protein 157 Is an Independently Evolved Inhibitor of Protein Kinase R"

_viruses, 2022, doi:10.3390/v14071564_

Round 1

Reviewer 1 Report

Comments for the Authors,

In this work, the authors reported a PKR inhibitor protein CRV157 from the crocodilepox virus. The CRV157 was first evaluated using bioinformatic analyses. Subsequently, recombinant vaccinia viruses with either CRV157 or a mutant CRV157 deficient in dsRNA binding were constructed, in which the vaccinia virus lacks PKR inhibitors. This study showed the ability of CRV157 to inhibit PKR correlated with virus replication and eIF2α phosphorylation. The methods and the data presented look convincing and well-presented. No issues are needed to be addressed before publication.

Reviewer 2 Report

In this manuscript the authors have characterized Crocodile pox virus protein 157 as an inhibitor of Protein kinase R. CRV157 can bind dsRNA, inhibit PKRs from different species, and its dsRNA binding ability is required for PKR inhibition. However, complete characterization of CRV157 function requires several experiments. The authors should perform few more experiments to confirm CRV157 as a potent inhibitor of PKR.

Other major concerns

1.    The author should indicate the residue numbers for dsRNA binding domain in Figure 4B

2.    For colocalization assay, the authors have used C-terminal tag. They must evaluate whether the tag has any influence on the localization by changing the tag or changing the position of the tag or antibody staining.

3.    CRV157 has been shown to bind to RNA, does it inhibit other RNA sensors? They can discuss this in the discussion section of the manuscript.

4.    Pox virus is a DNA virus, did the author check any DNA binding activity for CRV157.

5.    The authors have performed pull down experiment with whole cell lysates. Did the authors try to purify the CRV157 protein and check the interaction with nucleic acid with other biophysical techniques.

6.    K52 of CRV157 is also conserved (figure 2). The author should check its influence on RNA binding

7.    The author can represent the multiple sequence alignment in Figure 2 in a better way. They can use jalview. Why the VACV E3 K171 is shown in black? It should be colored in red.

Minor concern

1.    The authors should check the citations carefully. There are lots of formatting issues i. e. page 9 line 325

2.    The authors should check the manuscript carefully, there are several grammatical mistakes throughout the manuscript. i. e. line 450

Round 2

Reviewer 2 Report

The answers to some of the comments are not justified. Specially comment 6, their argument for not doing the experiment is not convincing.

Author Response

We are surprised and disappointed that our first revision, in which we addressed Reviewer 2’s comments in good faith, has somehow been reviewed more harshly than our initial submission. Moreover, Reviewer 2 has neglected to provide critiques outlining why their opinion of this manuscript has declined, making it difficult if not impossible to respond to the concerns.

Given no alternative, we will address the points that we can.

Reviewer 2 marked that “Extensive editing of English language and style [is] required”. As we commented in our initial response, this manuscript has been reviewed by two native English speakers, one of whom is a grant writing specialist and both of whom have extensive manuscript and grant writing experience. Thus, we vehemently disagree with Reviewer 2 on this point.

Reviewer 2 also said that our “…argument for not doing [the experiment suggested in comment 6] is not justified.” However, Reviewer 2 again neglects to elaborate on why she/he believes our response is not justified. Residues K48 and K49 have an extensive body of literature supporting their importance for dsRNA binding in multiple proteins [references 40,48,55,56,57]. While K52 is somewhat conserved in other dsRNA binding proteins, and might also play a role in dsRNA binding as other residues in CRV157 likely do, the purpose of generating the CRV157-K48A/K49A mutant was to have a control that is unable to bind dsRNA and not to extensively study all residues in CRV157 that play a role in dsRNA binding. Therefore, in our view investigating residue 52 is not necessary for an initial discovery paper and cannot justify the substantial time and expense to perform this experiment.